# Experimental and observational studies find contrasting responses of soil nutrients to climate change

ZY Yuan[1,2]*[†], F Jiao[1,2]*[†], XR Shi[1,2], Jordi Sardans[3,4], Fernando T Maestre[5], Manuel Delgado-Baquerizo[5,6], Peter B Reich[7,8], Josep Peñuelas[3,4]

[1]State Key Laboratory of Soil Erosion and Dryland Farming on the Loess Plateau, Institute of Soil and Water Conservation, Northwest A&F University, Yangling, China; [2]Institute of Soil and Water Conservation, Chinese Academy of Science and Ministry of Water Resource, Yangling, China; [3]Global Ecology Unit CREAF-CSIC-UAB, Consejo Superior de Investigaciones Científicas (CSIC), Bellaterra, Spain; [4]CREAF, Cerdanyola del Vallès, Spain; [5]Departamento de Biología y Geología, Física y Química Inorgánica, Universidad Rey Juan Carlos, Móstoles, Spain; [6]Cooperative Institute for Research in Environmental Sciences, University of Colorado, Boulder, Colorado; [7]Hawkesbury Institute for the Environment, University of Western Sydney, Penrith, Australia; [8]Department of Forest Resources, University of Minnesota, Minnesota, United States

*For correspondence: zyyuan@ ms.iswc.ac.cn (ZYY); jiaof@ms.iswc. ac.cn (FJ)

[†]These authors contributed equally to this work

Competing interests: The authors declare that no competing interests exist.

**Abstract** Manipulative experiments and observations along environmental gradients, the two most common approaches to evaluate the impacts of climate change on nutrient cycling, are generally assumed to produce similar results, but this assumption has rarely been tested. We did so by conducting a meta-analysis and found that soil nutrients responded differentially to drivers of climate change depending on the approach considered. Soil carbon, nitrogen, and phosphorus concentrations generally decreased with water addition in manipulative experiments but increased with annual precipitation along environmental gradients. Different patterns were also observed between warming experiments and temperature gradients. Our findings provide evidence of inconsistent results and suggest that manipulative experiments may be better predictors of the causal impacts of short-term (months to years) climate change on soil nutrients but environmental gradients may provide better information for long-term correlations (centuries to millennia) between these nutrients and climatic features. Ecosystem models should consequently incorporate both experimental and observational data to properly assess the impacts of climate change on nutrient cycling.

## Introduction

Shifts in patterns of precipitation and increases in temperature are two major components of ongoing climate change (**IPCC, 2014**). Manipulative experiments (e.g. field/mesocosm) and observational studies constitute the main empirical approaches typically used to forecast ecological responses to climate change (**Beier et al., 2012**; **Kreyling and Beier, 2013**; **Michelsen et al., 2012**). Both can have replication and randomization, but they are clearly distinct (**Baldi and Moore, 2014**; **Snecdecor and Cochran, 1989**). The major difference between these approaches is likely the issue of association versus causation; observational studies randomly select a sample of subjects and may find correlations between variables (**Rosenbaum, 2010**; **Yuan et al., 2016**). In comparison,

experimental studies can identify and confirm the potential mechanisms underlying observed responses by controlling any variable (*Montgomery, 2008*) and can assess cause-effect relationships based on the direct effects of climate change on small-scale, rapid, and short-term ecological processes (*Chapin and Shaver, 1996*; *Tilman, 1989*). If an experimental study is well-controlled, the factor that causes the difference can be reliably identified. In contrast, all confounding factors cannot be ruled out in observational studies.

Experimental responses are generally assumed to mimic long-term responses to climate change (*Luo and Hui, 2009*). This assumption may be due to the similar positive responses of biomass production to both experimental and spatial gradients of increasing temperature and water availability (*Lin et al., 2010*; *Lu et al., 2013*; *Rustad et al., 2001*; *Wu et al., 2011*). Recent studies, however, challenge this critical but largely unexplored assumption (*Blume-Werry et al., 2016*; *Metz and Tielboerger, 2016*; *Primack et al., 2015*; *Wolkovich et al., 2012*). Indeed, the direction and magnitude of observed impacts of climate change on ecosystems largely depend on whether the data are observational or experimental (*Metz and Tielboerger, 2016*; *Osmond et al., 2004*; *Sternberg et al., 2011*; *Wolkovich et al., 2012*), and these discrepancies have led to debates on the relevance of results of manipulative experiments for natural communities (*Hewitt et al., 2007*; *Sagarin and Pauchard, 2012*). Understanding the mechanisms behind the contrasting findings of observational and experimental approaches is thus essential to accurately forecast the impacts of climate change on the structure and functioning of ecosystems from local to global scales.

Soils are intricately linked to the atmospheric climate system through biogeochemical and hydrological cycles (*Chapin et al., 2011*; *Jarvie et al., 2012*; *Williams, 1987*), hence any change in climate is expected to influence soil characteristics and vice versa. Research in the last two decades has suggested various ways in which climate change will impact soil processes and properties, with critical consequences for the biogeochemical cycles of carbon (C), nitrogen (N), and phosphorus (P) (*Field et al., 2007*; *Hungate et al., 2003*; *Thornton et al., 2009*; *van Groenigen et al., 2006*). The direction and magnitude of these impacts largely depend on changes in precipitation and temperature, which are key drivers of biogeochemical cycles. The results from these studies, however, may vary greatly depending on whether the data are from observational or experimental studies (*Metz and Tielboerger, 2016*; *Osmond et al., 2004*; *Sternberg et al., 2011*; *Wolkovich et al., 2012*). Determining the nature of the responses of nutrient cycling to drivers of climate change identified by the two approaches is essential to accurately forecast the impacts of climate change on the availability of soil C, N, and P from local to global scales, and thus their potential impacts on ecosystem structure and functioning.

Temporal or spatial observations along environmental gradients at regional or global scales can provide important insights into the relationships between ecosystem processes and climate (*Dunne et al., 2004*; *Sagarin and Pauchard, 2012*; *Sternberg and Yakir, 2015*). Little information from long-term historical observations of soil nutrients is available, but spatial data across natural gradients of precipitation or temperature are abundant, offering an invaluable source for understanding the impacts of climate change on natural ecosystems at long-term scales. Results from both experiments and natural observations indicate that climate change largely influences key ecosystem processes, including those involving soil nutrients (*Elmendorf et al., 2015*; *Sternberg and Yakir, 2015*; *Wolkovich et al., 2012*; *Yuan and Chen, 2015a*). Much less is known, however, about the agreement between experimental and observational approaches for the direction and magnitude of the responses of nutrient cycling to climate change. In effect, differences in approaches are often theoretically recognized (*De Boeck et al., 2015*; *Dunne et al., 2004*; *Jiang et al., 2009*) but have rarely been empirically evaluated and quantified.

Biodiversity–productivity relationships contain an apparent paradox: experiments that directly manipulate species diversity often report a positive effect of diversity on productivity, whereas observations of natural communities identify various productivity–diversity relationships (*Jiang et al., 2009*; *Wardle and Jonsson, 2010*). Short-term experiments of N fertilization reported nearly ubiquitous negative productivity–diversity relationships (*Suding et al., 2005*). Contrasting patterns may be found for the diversity–invasibility relationship: experiments usually detect negative relationships, and field surveys find positive relationships (*Clark et al., 2013*; *Fridley et al., 2007*). The paradox has led to debates on the relevance of the results of manipulative experiments for natural communities, and understanding the mechanisms behind the apparent conflicts is therefore a major step toward developing general theories of community processes, including those for soil nutrients.

Nutrient cycling plays a fundamental role in ecosystem services such as food and fiber production, C sequestration, and climate regulation, so this knowledge gap may hamper our ability to accurately predict ecosystem functioning under climate change. Fortunately, compared with the lack of data for other ecosystem response variables (e.g. species diversity, gas exchange, and microbial attributes), a wealth of data for soil nutrients is available for testing the consistency between experimental and observational approaches.

No study, to the best of our knowledge, has directly tested the congruence of soil nutrient responses to changing regimes of precipitation or temperature obtained from experimental *vs.* observational approaches, despite the importance of soil for ecosystem functioning. We thus conducted a meta-analysis of 1421 data points from 182 experimental studies and of a total of 1346 sites from 141 studies of natural gradients around the world (see *Figure 1* and Supplementary References). We evaluated the similarity of the predictions from experimental and observational approaches by assessing the direction of the responses of soil nutrients to changing precipitation/ temperature. Further analyses were conducted to compare the response ratios from experimental studies to changes in precipitation or temperature at the various study sites with the overall response across all observational studies as a function of precipitation, aridity and temperature. We used total nutrient concentrations because they represent long-term nutrient reservoirs in terrestrial ecosystems and are more stable than extractable nutrients in soils. Total C, N, and P concentrations generally indicate overall nutrient availability and are positively correlated with each other (*Bertiller et al., 2006*; *Jacoby, 2005*; *Silver, 1994*). Given that temperature varies across precipitation gradients we also used aridity, which incorporates both potential evapotranspiration and precipitation.

We hypothesized that short-term local manipulative experiments and large-scale observational studies identify different responses of soil nutrients to changes in climatic conditions. These differences may be due to the manner in which the entire ecosystem responds to rapid (i.e. days to months, e.g. drought) *vs.* slow (i.e. centuries to thousands of years, e.g. changes in aridity) climatic changes. The properties of ecosystems thus often respond in a simultaneous and predictive manner to slow changes in climate (e.g. plant cover, diversity, soil properties and nutrient pools), but rapid climatic

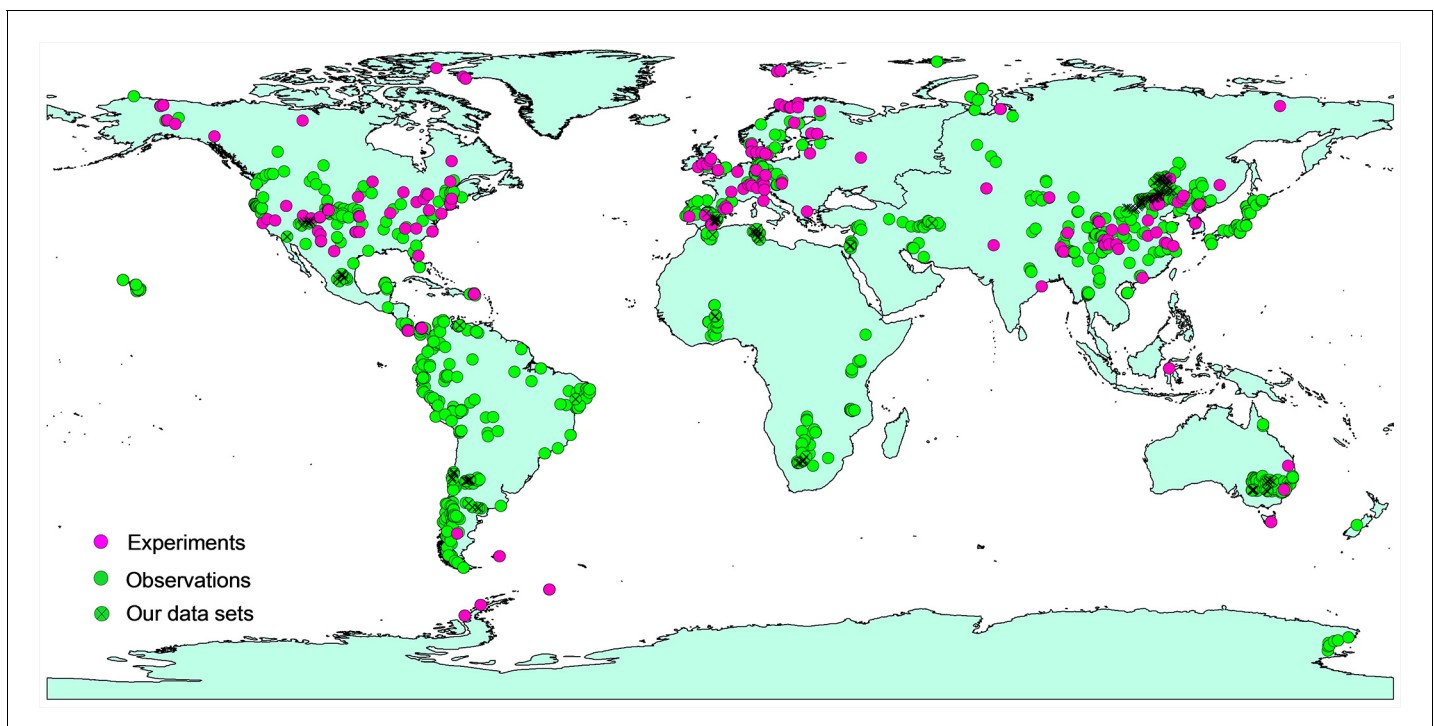

**Figure 1.** Map of the experimental and observational sites used in our analyses.

changes will likely disrupt the simultaneous responses of these slowly changing processes to climate change. For example, a change in water supply in manipulative precipitation experiments in local environments (e.g. similar plant species, plant cover, or soil type) could directly influence the balance between nutrient uptake/leaching *vs.* mineralization/decomposition, particularly in arid ecosystems (*Ansley et al., 2014*; *Davidson et al., 2004*). In contrast, and in addition to precipitation, abiotic factors (such as temperature and soil age and taxonomy) and vegetation co-vary simultaneously along a precipitation gradient. For example, increases in aridity across a gradient often lead to reductions in the diversity and cover of vascular plants, rates of litter decomposition, and availability of C and N (*Delgado-Baquerizo et al., 2013*; *Jiao et al., 2016*), all of which co-evolve over millennia (*Ehrenfeld et al., 2005*; *Lambers et al., 2009*). All these co-varying factors likely affect the processes involved in the cycling and storage of soil C, N, and P. We thus posit that short-term local manipulative experiments could provide useful information for predicting shifts in soil processes in response to rapid changes in climate (from years to decades, e.g. drought), such as those we are facing today. In contrast, observational studies might provide unique information for understanding the response of soil processes to long-term changes in climate (from centuries to thousands of years, e.g. increases in aridity [*Huang et al., 2015*]).

As stated above, the contrasting findings between manipulative experiments and observational studies may be due to the timescale, because the former have typically been performed at short temporal scales in relatively homogeneous environments to maximize experimental control, whereas the latter have usually been established over largely spatial gradients that reflect long-term adjustments of ecosystems to local climate (*Kueppers and Harte, 2005*). Given that the availabilities of soil nutrients reflect both short- and long-term changes in climate, soil and ecosystem development (*Vitousek et al., 2010*), we hypothesized, therefore, that experimental and observational studies would not produce similar results for soil nutrients in response to climate change.

## Results

Contrasting responses of soil total C, N, and P concentrations were found between experimental and observational studies. The responses of soil total C, N, and P concentrations generally differed between paired manipulative experiments and environmental gradients (*Figures 2–4*, *Figure 2—figure supplement 1*). Overall, adding water in experiments decreased soil N, P, and C concentrations by 5.4, 9.3, and 2.7%, respectively (non-significantly for C). In contrast, N concentration increased with precipitation across environmental gradients. The enhanced effects of increasing precipitation on C and P concentrations were not significant (*Figure 2*). Experimental drought treatments significantly increased the concentrations of C, N, and P by 6.1, 17.9, and 6.8%, respectively. Soil N and P concentrations decreased with increasing aridity and C concentration did not change significantly when precipitation was replaced by aridity. The aridity-related patterns were opposite to those observed in manipulative drought experiments (*Figure 3*).

Responses to changes in temperature also differed between experimental and observational studies. Experimental warming generally increased the concentrations of C, N, and P, especially that of N (*Figure 4*). In contrast, these concentrations tended to decrease along gradients of increasing mean annual temperature (*Figure 4*). Similar results were found for temperature-related changes in C, N, and P concentrations across different types of ecosystems or soils: patterns were inconsistent between manipulative experiments and environmental gradient observations for the same climate-change factor of precipitation or temperature (*Figures 2–4*).

Cross-study analyses found that response ratios varied between experimental studies along climatic gradients. The response ratio of soil C concentration increased quadratically with mean annual precipitation in water-addition experiments, but decreased non-significantly with precipitation in drought experiments (*Figure 5*). The response ratios of N and P concentrations in manipulative precipitation experiments were not significantly correlated with mean annual precipitation. The response ratio of P concentration in drought experiments decreased with precipitation (*Figure 5*). This relationship, however, was mainly dependent on two data points (*Figure 5*) and disappeared when they were removed ($R^2$ = 0.01, p=0.84). The response ratio of C concentration in warming experiments was linearly correlated with mean annual temperature. Unimodal curves best described the relationships between the response ratio of N and P concentrations in warming experiments and

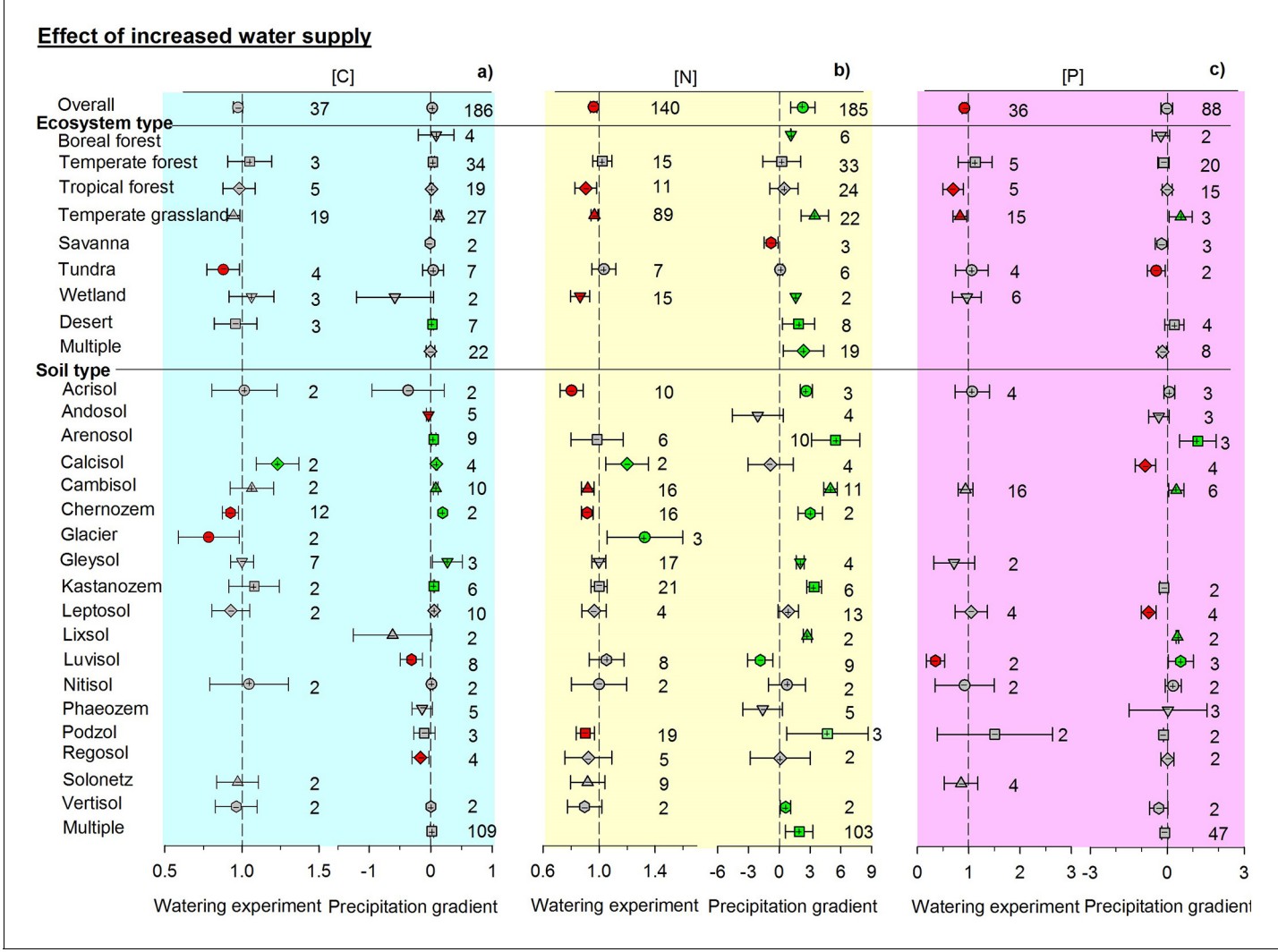

**Figure 2.** Responses of soil (**a**) C, (**b**) N, and (**c**) P concentrations to wetting by using response ratios in manipulative experiments and slopes in precipitation gradient observations. Symbols with error bars show the mean response ratios or slopes with 95% confidence intervals. Plus (+) and minus (–) signs represent positive and negative means, respectively. Green symbols are positive and red symbols are negative means whose confidence intervals do not include zero. 'Multiple' indicates results from studies conducted across an environmental gradient for multiple types of ecosystems or soils. Manipulative experiments are on the left and gradient observations are on the right of each panel. Numbers indicate the number of studies used in each case.

The following figure supplement is available for figure 2:

**Figure supplement 2.** Response of soil C:N, C:P and N:P to precipitation, temperature and aridity in manipulative experiments (**a**) and gradient observations (**b**).

mean annual temperature. At low to moderate levels, mean annual temperature increased the response ratio of P concentrations.

In the cross-study analyses of all observational studies, climatic data (temperature and precipitation) collectively explained ~30% of the variation in soil total C, N, and P concentrations. Soil and ecosystem types were also strong drivers of total C, N, and P concentrations. When all data were included, the variables of climate, aridity, and type of ecosystem and soil accounted for ~63–77% of the variation observed in the C, N, and P concentrations (*Table 1*). Soil C, N and P concentrations, based on all observations from the observational studies, changed quadratically with mean annual precipitation and temperature (*Figure 6*). Both C and N concentration were significantly and

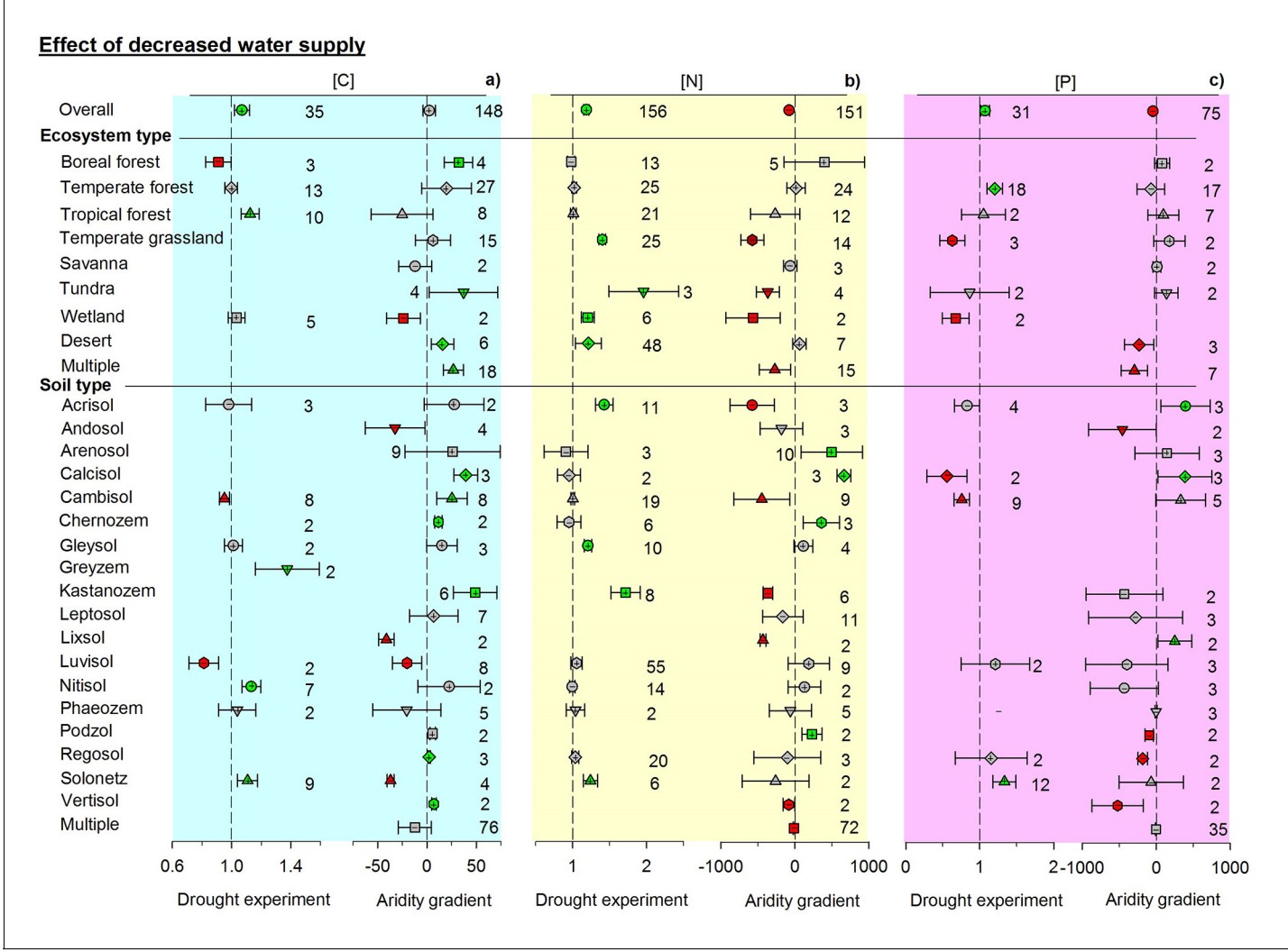

**Figure 3.** Responses of soil (a) C, (b) N, and (c) P concentrations to drying by using response ratios in manipulative drought experiments and slopes in aridity gradient observations. Aridity is defined as [maximum AI in the data set – AI], where AI is the aridity index, the ratio of precipitation to potential evapotranspiration. Same symbols and explanations as in *Figure 2*.

positively correlated with mean annual precipitation but negatively correlated to aridity. Soil P was negatively correlated with precipitation but was not significantly correlated with the aridity. Soil C, N and P were quadratically correlated to with temperature. Similar but clearer patterns were found between soil nutrients and climate variables when only considering the controls from all experiments (*Figure 7*). Soil P concentrations increased at low level of precipitation but decreased at high level of precipitation. Neither C nor N were significantly correlated with precipitation. In contrast, soil C, especially N and P, significantly decreased with aridity. Soil C and N were negatively correlated with temperature but soil P was positively correlated with temperature.

## Discussion

Our findings provide evidence that manipulative experiments and observational studies conducted along environmental gradients identify contrasting correlations of soil C, N, and P concentrations with changes in precipitation and temperature. Data from short-term manipulative experiments may therefore not always agree with long-term gradient data. The contrasting results observed may be the consequence of the timescale that applies to local and large scales in terrestrial ecosystems. Our

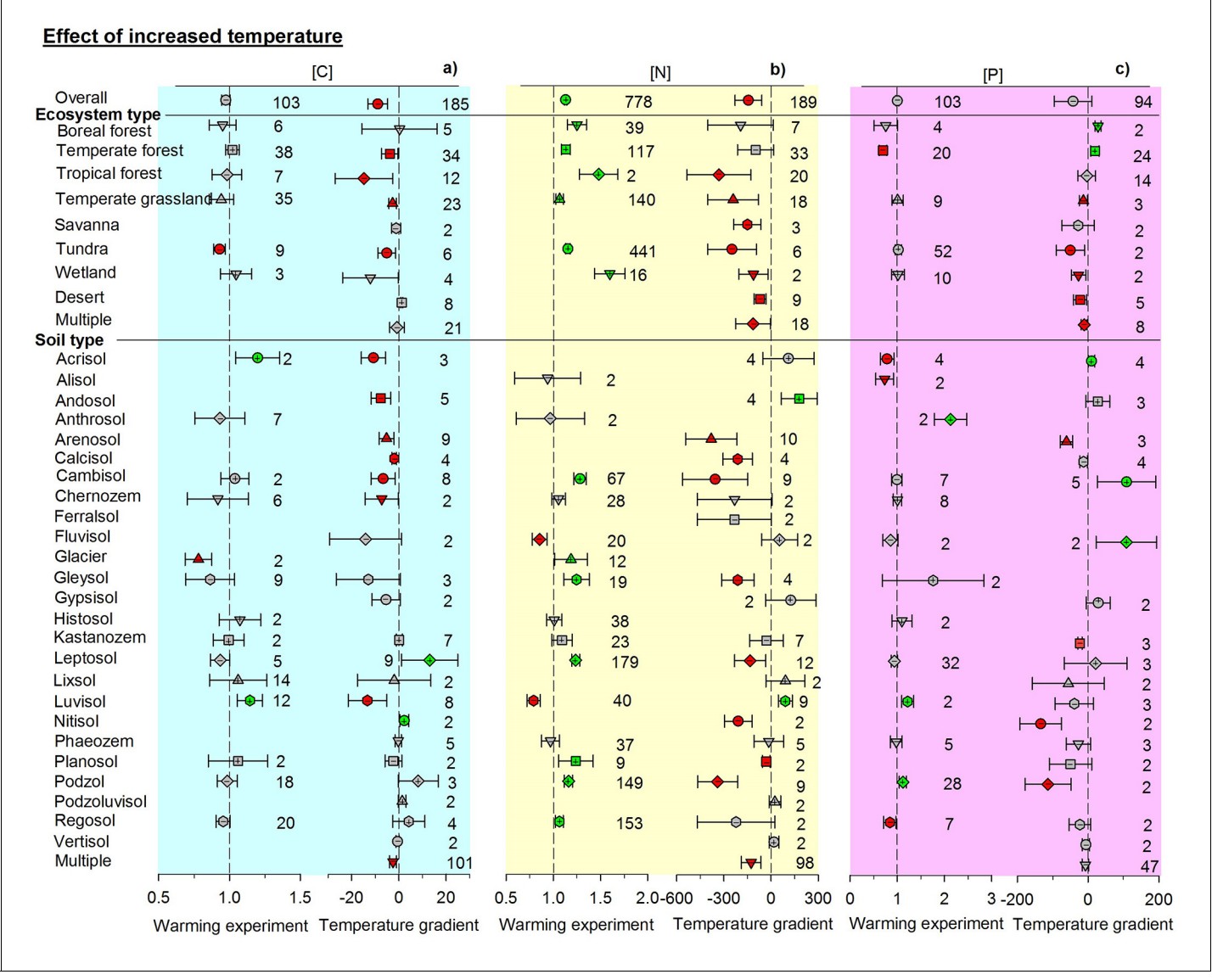

**Figure 4.** Responses of soil (**a**) C, (**b**) N, and (**c**) P concentrations to warming by using response ratios in manipulative temperature experiments and slopes in temperature gradient observations. Same symbols and explanations as in *Figure 2*.

analyses suggest that soil nutrients should respond differently to experimental short-term warming and drought, but if climatic conditions continue to change, the responses might progressively become more similar to those derived from environmental gradients due to ecosystem adaptation to climate change and to the co-evolution of multiple soil properties and plant features (*Neil Adger et al., 2005*). Our analyses highlight the need to further explore the underlying mechanisms of the inconsistent patterns in the two approaches and to determine the scenarios under which each is more likely to provide relevant insights when assessing the responses of ecosystems to climate change.

It is increasingly clear that precipitation provoked by climate change will alter nutrient cycles, although these responses induced by rainfall may vary among ecosystems and soils (*Knapp et al., 2008*; *Sardans and Penuelas, 2007*; *Yuan and Chen, 2015b*). The increased water availability with higher rainfall boosts microbial activity and plant-nutrient uptake, and hence nutrient cycling. In contrast, a drier climate reduces microbial nutrient uptake but increases the availability of soil nutrient, leading to an accumulation of nutrients. This higher nutrient pool in soil, however, may increase the

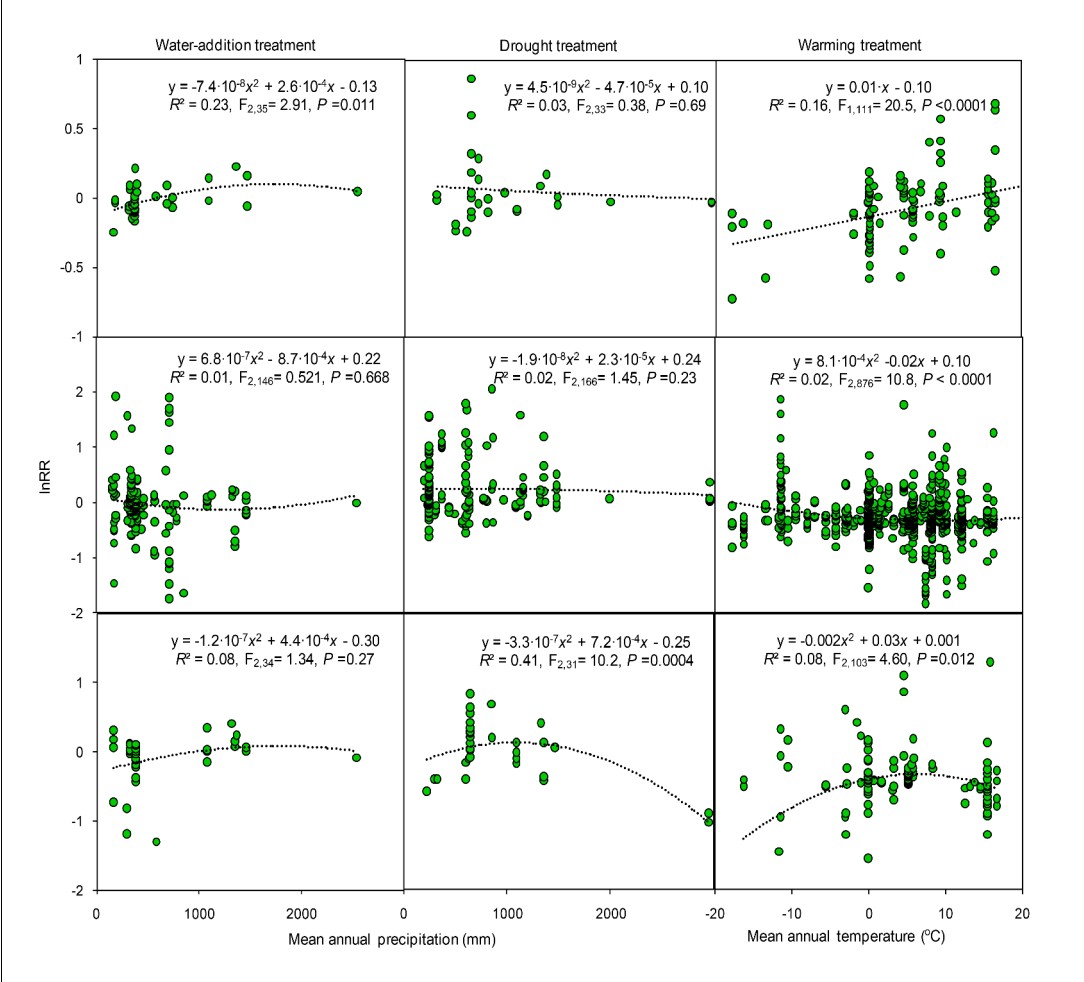

**Figure 5.** Results of regression analysis for the response ratio (lnRR) in relation to mean annual precipitation and temperature in manipulative experiments of study sites. Details of the fitted models are given within each panel.

risk of nutrient loss by leaching or erosion, leading to a short- to middle-term nutrient impoverish-ment (*Matias et al., 2011*; *Ramos and Martínez-Casasnovas, 2004*). The decrease in soil N and P concentrations we found in manipulative precipitation experiments can thus be attributed to changes in the balance between nutrient inputs (e.g. litter decomposition) and outputs (e.g. leaching and emission of gases to the atmosphere) (*Austin et al., 2004*; *Carrillo et al., 2012*; *Cleveland et al., 2010*; *Yahdjian et al., 2006*). For example, experimentally elevated precipitation significantly increased $N_2O$ fluxes by volatilization and denitrification in an annual grassland (*Niboyet et al., 2011*), especially in combination with N addition and warming (*Brown et al., 2012*). Moreover, adding water can stimulate plant growth, particularly in drylands, and can thereby lead to increased sequestration of nutrients in plant biomass, which could be another mechanism for the decrease in soil N and P concentrations observed in water-addition experiments.

In contrast, gradients of increasing precipitation are associated with an increase in soil total N. Accumulation/decomposition of organic matter, hence the import/export of 'new/old' organic mat-ter, are controlled by many climatic factors, soil physical and chemical conditions, and vegetation type and productivity. All these factors vary simultaneously across precipitation gradients, and prob-ably contribute to the observed inconsistent pattern when compared to experimental observations. Increasing aridity often leads to a strong reduction in plant cover (*Vicente-Serrano et al., 2013*), leading to reduced inputs of plant litter and rates of decomposition, so less N is released. Moreover, increases in aridity often lead to strong increases in sandy fractions, reducing the capacity of soil to

**Table 1.** $R^2$ values of multiple regression analyses of soil carbon, nitrogen, and phosphorus concentrations for the observational studies. Abbreviations: T, mean annual temperature; P, mean annual precipitation; A, aridity; S, FAO soil classification; E, type of ecosystem. Letter combinations indicate which explanatory terms are included in a model. Values in bold indicate significant effects (p<0.05).

|  | [C] | [N] | [P] |
|---|---|---|---|
| T | 0.051 | 0.012 | 0.018 |
| P | 0.222 | 0.061 | 0.007 |
| A | 0.319 | 0.181 | 0.001 |
| S | 0.334 | 0.179 | 0.107 |
| E | 0.420 | 0.215 | 0.158 |
| TP | 0.342 | 0.190 | 0.035 |
| TA | 0.331 | 0.201 | 0.070 |
| TS | 0.494 | 0.252 | 0.248 |
| TE | 0.509 | 0.288 | 0.285 |
| PA | 0.386 | 0.213 | 0.018 |
| PS | 0.562 | 0.230 | 0.166 |
| PE | 0.510 | 0.270 | 0.156 |
| AS | 0.492 | 0.349 | 0.212 |
| AE | 0.514 | 0.340 | 0.261 |
| SE | 0.700 | 0.350 | 0.359 |
| TPA | 0.424 | 0.285 | 0.115 |
| TPS | 0.685 | 0.349 | 0.322 |
| TPE | 0.583 | 0.358 | 0.322 |
| TAS | 0.569 | 0.450 | 0.361 |
| TAE | 0.566 | 0.446 | 0.374 |
| TSE | 0.758 | 0.411 | 0.484 |
| PAS | 0.564 | 0.418 | 0.256 |
| PAE | 0.577 | 0.432 | 0.307 |
| PSE | 0.743 | 0.409 | 0.418 |
| ASE | 0.671 | 0.554 | 0.479 |
| TPAS | 0.621 | 0.511 | 0.469 |
| TPAE | 0.617 | 0.542 | 0.434 |
| TPSE | 0.799 | 0.493 | 0.562 |
| TASE | 0.730 | 0.668 | 0.582 |
| PASE | 0.726 | 0.632 | 0.542 |
| TPASE | 0.771 | 0.727 | 0.629 |

retain organic matter. N losses can thus be important both in wet (mainly by leaching) and dry (mainly hydrologic losses and atmospheric emission, especially fire-induced N losses by denitrification) sites (*Austin and Sala, 2002*; *Galloway et al., 2003*; *McCulley et al., 2009*), leading to lower levels of soil C and N and to a lesser extent of P in drier systems (*Delgado-Baquerizo et al., 2013*). Dry sites often fix substantial amounts of N (cyanobacteria are actually promoted by these conditions), but total N decreases with aridity because of the strong reduction in plant cover (hence in litter production and decomposition) and the common increase in coarse textured fractions (together with fewer plants and more sand, leading to higher N leaching in response to extreme events).

Interestingly, P concentrations responded less than N concentrations to changes in precipitation and temperature. Many reasons may account for this disconnection between climate and soil P

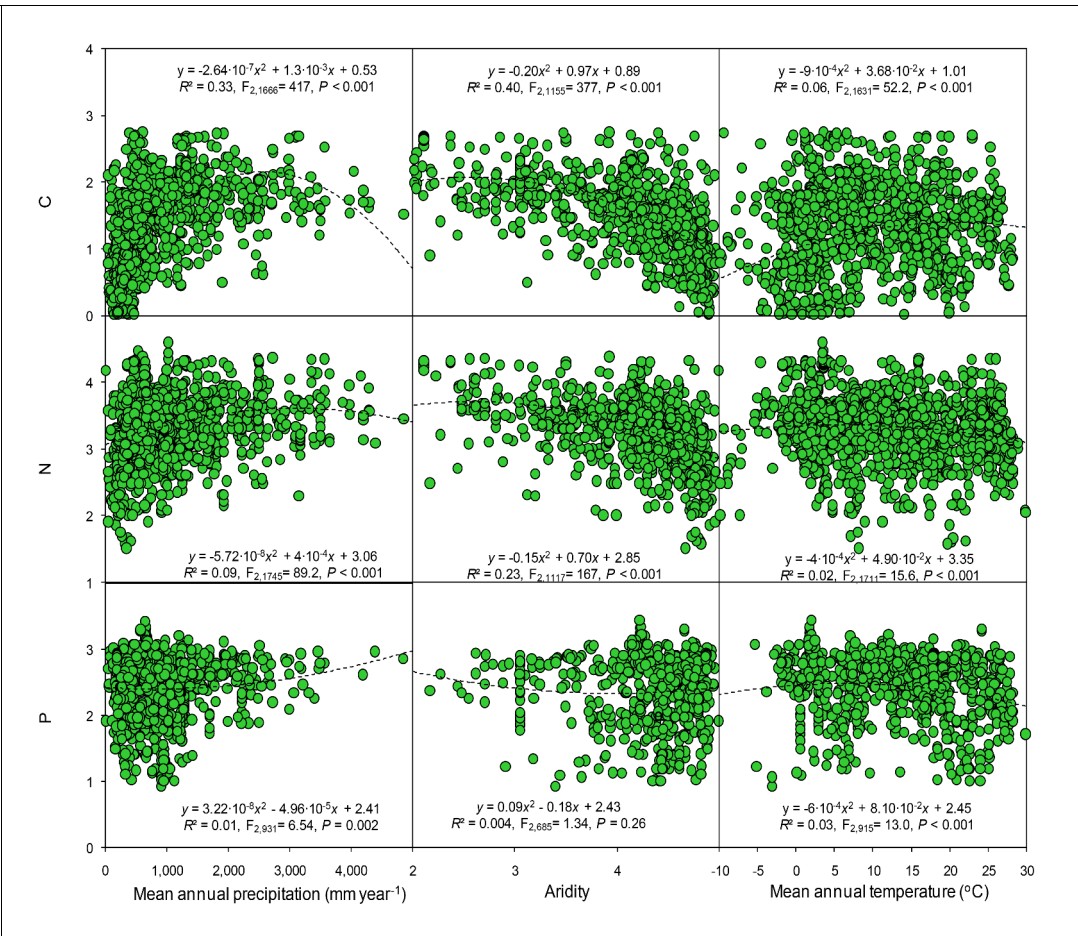

**Figure 6.** Results of regression analysis for the soil carbon, nitrogen, and phosphorus concentrations in relation to gradients of mean annual precipitation, aridity and temperature in observational studies. Details of the fitted models are given within each panel. Aridity is defined as [maximum AI in the data set – AI], where AI is the aridity index, the ratio of precipitation to potential evapotranspiration.

across environmental gradients. For example, soil P is strongly linked to soil type and bedrock availability. Two sites with a similar climate may thus have a naturally different amount of soil total P. Soil P is also highly influenced by soil age. P is thus highly limited in ancient soils, primarily caused by P depletion due to prolonged weathering over millennia, even though soil P is often bound in complexes with Al, Ca, Fe, or allophane clay in young soils that contain an abundance of total P but where P is poorly available (*Vitousek et al., 2010*; *Walker and Syers, 1976*). We did not control for soil age in our study, which may also account for the noisy relationship between soil P and climatic gradients. In addition, vegetation type may strongly influence the availability of P in soils. For example, in P-limited ecosystems, whether ancient, P-impoverished soils or young, potentially fertile soils (rich in total P, but with low P availability), plant species (e.g. Proteaceae family in Austral an soils) with cluster roots, cluster-like roots, or dauciform roots that exude large amounts of P-mobilizing carboxylates (organic anions) may promote P mobilization and thus act as engineers and/or facilitators in ecosystems by converting poorly available P to more readily available forms and by enhancing P uptake of neighboring plants (*Lambers et al., 2012*; *Plaxton and Lambers, 2015*). The lack of explicit control of the identity of plant species in our analyses may explain part of the noise in the soil P–climate relationship. The finding that the concentration of soil P was less affected by climate than those of C and N that are strongly controlled by the activity of organisms, and hence by climate, was not surprising, for the reasons presented above.

Differences between observational and experimental studies evaluating biodiversity–productivity relationships could be understandable because the artificial selection of component species in short-

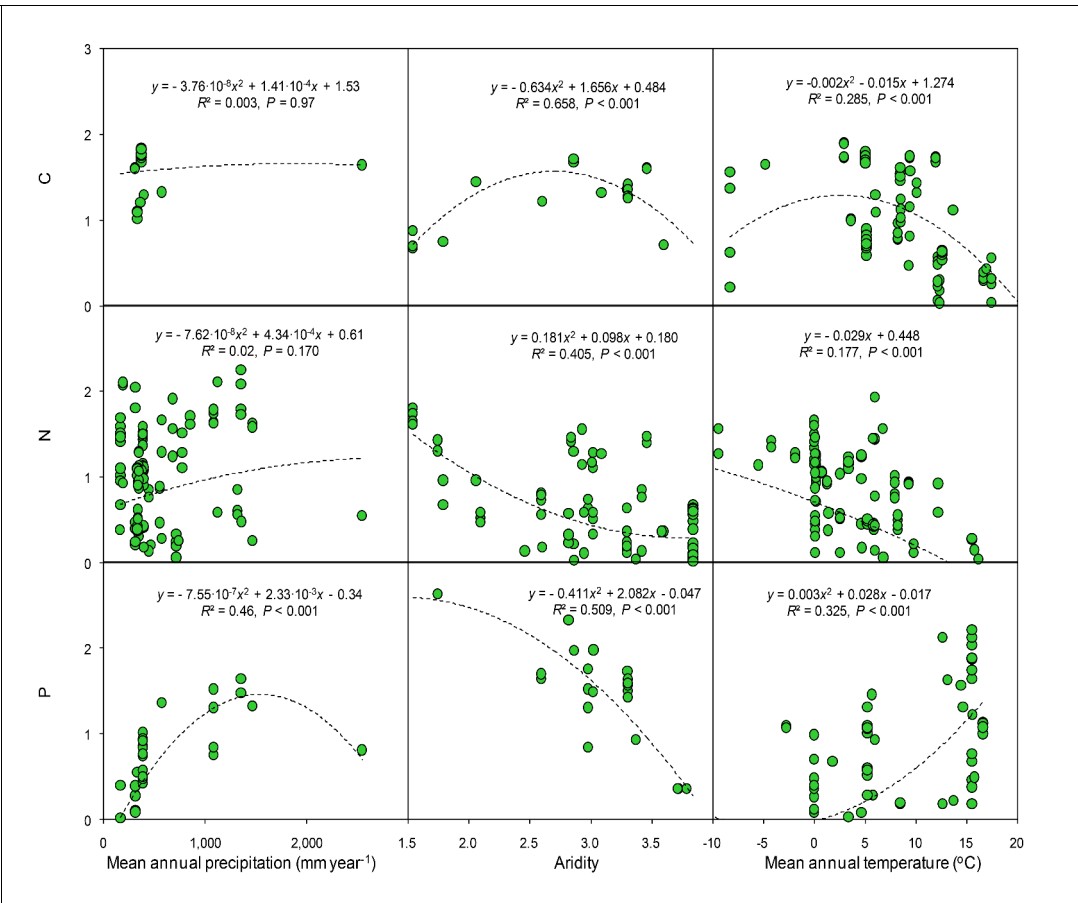

**Figure 7.** Results of regression analyses for the soil carbon, nitrogen, and phosphorus concentrations with gradients of mean annual precipitation, aridity and temperature using the controls from experiments. Details of the fitted models are given within each panel. Aridity is defined as [maximum AI in the data set – AI], where AI is the aridity index, the ratio of precipitation to potential evapotranspiration.

term manipulative experiments can lead to a 'sampling' or 'positive selection' effect, whereby randomly selecting a more productive species is more likely in more diverse treatments (*Tilman et al., 1996*). Similarly, the contrasting patterns we found between observational and experimental studies might be because climate change tends to have a direct and simple effect on soil nutrients in the relatively homogenous environments that characterize small-scale manipulative experiments conducted at a single field site, but the direct and simple effect tends to be overwhelmed by the environmental heterogeneity (e.g. spatial variation in soil nutrient concentration) that characterizes large-scale observational studies. The hypothesis of environmental heterogeneity proposed in conceptual models for biodiversity–productivity relationships (*Hector et al., 2007*) and the differences in spatial scale driving the invasion paradox (*Fridley et al., 2007*) could also possibly explain the contrasting patterns observed in our study.

The responses of nutrient cycling to temperature, as with precipitation, differed between experimental and observational approaches. The increased soil N concentration but non-significant response of P concentration observed in warming experiments may be due to sustained increases in net N mineralization and nitrification (*Butler et al., 2012*; *Melillo et al., 2011*) with warming observed in diverse ecosystems. Our meta-analysis of soil nutrients indicated that experimental warming increased the N:P ratio by 8.6% (*Yuan and Chen, 2015a*), again suggesting different responses of N and P concentrations to climate change. Biological activity in an ecosystem, especially in temperate areas, increases under warming, which can increase soil N stocks (due to more N fixation) and uptake by microbes. Similar responses are more difficult for P, because its concentration in soil is less regulated by biological processes. Increases in temperature in drylands, however,

are often associated with reductions in water availability, which can promote N limitation due to impaired microbial N cycling in soil (*Hooper and Johnson, 1999*).

Timescale is likely a key reason for the differences between observational and experimental studies. Experiments essentially measure the initial stages of the effect of a sudden change in climate on ecosystems (e.g. +3.5°C of temperature) so that slowly changing properties of ecosystems (e.g. plant cover and soil properties) do not have enough time to co-evolve with this climatic change. In contrast, observational field studies measure ecosystems with a history over an unknown period, perhaps decades, centuries, or even millennia or more. Field experiments suddenly submit one community to conditions different than those to which it is adapted, whereas observations compare different communities that have evolved in different conditions. The community is stressed by experiments but is on its optimal environment along gradients. We cannot therefore expect equal responses. Experimental studies thus provide information about the extent to which the vegetation can support a sudden change, whereas observational gradient studies provide information about evolutionary responses.

The difference between observational and experimental studies may diminish if manipulative experiments are allowed to run for a sufficiently long time, because long-term interactions within communities may likely adjust soil nutrients towards those seen in natural communities. The time, however, should be of a magnitude that allows species substitutions in treatment plots to reach results similar to those along a comparable gradient, which would be impossible on human timescales. For example, if we apply a drought experiment to a beech forest in northern Germany and compare it with a level of drought equivalent to that of a semi-Mediterranean *Quercus ilex* forest in southern France, we must give sufficient time to the experimental drought plots to reach results similar to those of an observational gradient study to allow a *Q. ilex* community to replace the beech forest, which would not occur within a reasonable time for experimental studies. Most manipulative experiments are typically short-term, generally no more than a few growing seasons, so we do not know whether experimental studies of much longer duration would lead to response patterns that are more similar to those from observational studies. *Harte et al. (2015)* found that soil organic C in a 23 year warming experiment in montane grassland of Colorado Rockies declined by 25% during the first six years and increased thereafter until it reached approximately its preheating level in the 23th year of warming. Furthermore, observational studies may often not have been designed as climate change studies, allowing potential confounding effects to influence the results. Perhaps the most striking result would be the change of the vegetation composition along gradients (e.g. colonization by legumes could affect soil N irrespective of precipitation, drought or temperature). The effect on soil nutrients observed in manipulative experiments may thus represent the 'true' effect of rapid climate change, so experimental studies clearly produce more consistent results than observational studies. Notably, the responses of soil nutrients in short-term manipulative experiments within a single field may not necessarily apply to long-term manipulative experiments possibly due to an adjustment of the plant species to the altered environmental conditions that decrease the effect size in the long-term. Our findings thus highlight the importance of comparing short- and long-term effects side by side when forecasting the responses of an ecosystem to climate change.

In summary, our analyses suggest that experimental and observational approaches identify contrasting responses of soil nutrients to climate change. Manipulative experiments, likely indicating short-term responses (months to years) prior to coincidental shifts in plant and microbial compositions that could counteract short-term responses, may be better predictors of the near-term impacts of climate change on soil nutrients. Observations along spatial gradients may thus be more indicative of changes over longer timescales (centuries to millions of years) when multiple aspects of the ecosystem have had a chance to adjust. The responses of soil nutrients found in experimental studies may reflect a 'true' short-term and rapid effect of climate change, whereas spatial variation in environmental factors in large-scale gradient observations is likely to heterogeneously influence climate-nutrient relationships, thus supporting the hypothesis of environmental heterogeneity (*Dufour et al., 2006*; *Hector et al., 2007*) in explaining the discrepancy in climate–nutrient patterns between experimental and observational studies. These differences clearly alert us to the risk of misinterpreting short-term experimental results and long-term observations due to the different timescales involved in each of them, especially at broad geographical scales that are structured by multiple internal and external drivers. Manipulative experiments and environmental gradient observations are both valuable, but we still need to recognize the inferential limitations of these two commonly used

approaches and interpret their results cautiously. Experimental studies reproduce the conditions expected in the coming decades, thus simulating very fast rates of change that do not permit a shift in the distribution of vegetation, so experimental studies clearly produce more consistent results than observational studies. Experimental studies give us information about the extent to which the vegetation can support a sudden change whereas gradient observational studies give us information about the evolutionary responses of ecosystems to different conditions. Our study fills a critical knowledge gap and further suggests that both experimental and observational data are necessary to properly assess the responses of nutrient cycling to climate change.

## Materials and methods

The studies included in our meta-analysis were identified by searching the databases of the Institute for Scientific Information's Web of Science, PubMed, Google Scholar, and JSTOR for 1945–2015 using combinations of the following keywords: 'manipulative experiment', 'soil carbon/nitrogen/ phosphorus', 'climate change', and 'gradient' (a list of the selected studies is presented in the Supplementary References). Criteria for inclusion in our meta-analysis for manipulative experiments included (1) a report of at least one variable of soil C/N/P concentration in both manipulated and control groups, (2) the ability to calculate, the mean, standard deviation and sample size of reported C/N/P concentrations, and (3) the study was conducted in natural or semi-natural ecosystems (i.e. only non-agricultural ecosystems). The criteria for the gradient studies included (1) a report of differences in C/N/P concentration, and (2) data derived from field investigations, excluding data from studies in which environmental factors were manipulated or controlled. Only studies that reported total C, N, or P concentration using similar techniques of chemical analyses were included in our meta-analysis. We chose the data for which C concentration was determined by dichromate-acid digestion, N concentration was measured by the Kjeldahl method, and P concentration was determined colorimetrically (*Carter, 2007*).

In addition to collecting published data for the meta-analysis of observational studies, we collected and analyzed soil samples from a field observational study conducted at 65 sites along a 3500 km precipitation gradient in temperate grasslands of Inner Mongolia, China. These sites were in arid, semi-arid, and dry sub-humid areas covering a wide climatic spectrum. The average Mean Annual Temperature (MAT) across the 65 sites was 2.1°C, with a tendency for higher temperatures in the southwest. Mean Annual Precipitation (MAP) decreased from 457 to 154 mm $y^{-1}$, suggesting a transition from a humid to an arid climate. The corresponding summer rainfall (May to September) during the growing season varied from 404 to 133 mm $y^{-1}$. This precipitation gradient can also be viewed as an aridity or temperature gradient due to the close relationship among aridity, precipitation, and temperature in this region. We collected 15 soil samples from the 0–30 cm layer at each site with a soil corer (5 cm in diameter). The samples were taken to the laboratory, air-dried, and sieved for the analyses of C, N, and P concentrations. Organic-C concentration was determined colorimetrically following dichromate oxidation by boiling a mixture of potassium dichromate and sulfuric acid. Total N concentration was measured using Kjeldahl acid digestion. Total P concentration was determined by hydrolysis with sulfuric acid (*Carter, 2007*). We also used published and unpublished data from 259 dryland sites from all continents except Antarctica provided by two of our coauthors (Delgado-Baquerizo and Maestre). These included the 224 sites that were used in *Maestre et al., 2012* plus additional sites from Australia, Kenya, Botswana, Ghana, and Burkina Faso. Soil samples at these sites were collected as described by *Maestre et al. (2012)*. These original observational data were used in the same way as the meta-data collected from the literature. Our data thus covered a range of types of terrestrial ecosystems, including arctic tundra, forests, and grasslands. Forests were subdivided into boreal, temperate/subtropical, and tropical forests. Boreal forests included forests between 46 and 66°N, tropical forests included forests between 23.5°S and 23.5°N, and temperate/subtropical forests included forests between the tropical and boreal latitudes. Grasslands were also subdivided into temperate and tropical grasslands/savannas based on a latitudinal threshold of 23.5°N or S.

All meta-data were extracted from the text, tables, figures, and appendices of the original publications. When data were presented graphically, numerical data were extracted with Image-Pro Plus 7.0 (Media Cybernetics, Inc., Rockville, MD, USA). Measurements from different ecosystem types and treatment levels within a study were considered as independent observations. Different sites

within the same study have the same climate, so the results from different sites within each study are not independent. We deal with the lack of independence following the approach of *Vilà et al. (2011)* who compared the results of the analysis of the complete database with those from the analyses of a reduced database formed by one randomly selected data point per experimental study, or geographic location in the case of multisite studies. All data can be confidently included in the analysis if mean effect sizes are similar and the bias-corrected 95% bootstrap confidence interval (CI) overlaps between the complete and reduced databases. This approach has been successfully applied in many ecological meta-analyses (e.g. *Eldridge et al., 2011*; *García-Palacios et al., 2013*).

The data set included 182 experimental precipitation/warming studies and 141 studies (including the two own studies mentioned above) of natural precipitation/temperature gradients from around the world (Supplementary References). We used WorldClim (http://www.worldclim.org) (*Hijmans et al., 2005*) to get temperature/precipitation data for 1950–2000 using the R package '*raster*' based on the latitude and longitude of the sites. We used MAT and MAP in our analyses because they could be both derived from WorldClim, whereas climate data for individual growing seasons of various regions were not available. We also used aridity for our modeling, because aridity across large temperature gradients incorporates both potential evapotranspiration and precipitation and is thus more meaningful than precipitation alone. The aridity index (AI) is the ratio of MAP to mean annual potential evapotranspiration (MAE). MAP values were obtained from WorldClim. Potential evapotranspiration (PET) layers were estimated based on monthly averages from Global-PET (http://www.cgiar-csi.org) (*Zomer et al., 2008*) and were aggregated to obtain MAE. The AI decreases as aridity increases and is generally correlated with climatic characters such as rainfall and temperature. To facilitate the interpretation of our results, we substituted the AI by aridity, estimated as [maximum AI in the data set – AI], which increases with decreasing MAP ($R^2 = 0.68$, p<0.001). Aridity, as a negative substitute for AI, had an opposite relationship, i.e. AI was positively correlated with MAP, with the same $R^2$ and $P$ value. Aridity might be more appropriate than precipitation for the observational gradients in which temperature also varies. This index, however, is not suitable for manipulative experiments in which it does not vary accordingly. Given that the purpose of our analyses was to identify the similarities and differences of responses to precipitation and temperature between observational gradients and manipulative experiments, we chose precipitation over aridity for the comparison of experimental studies. Soil types, classified using the FAO-UNESCO classification system, were derived from the Harmonized World Soil Database version 1.21 released in 2012 by the FAO (*Batjes, 2009*).

In the case of manipulative experimental studies, we examined the effects of treatments of climate change on soil elements by calculating response ratios for each study as described by *Hedges et al. (1999)*. All nutrient ratios were log-transformed to achieve normality before conducting these analyses. The natural-log response ratio (lnRR) was calculated as $\ln(X_e/X_c) = \ln X_e - \ln X_c$, where $X_e$ and $X_c$ are the responses of each observation in the experimental treatment and the control, respectively. The corresponding sampling variance for each lnRR was calculated as $\ln[(1/n_e) \times (S_e/X_e)^2 + (1/n_c) \times (S_c/X_c)^2]$ using the R package '*metafor*' 1.9–8 (*Viechtbauer, 2010*), where $n_e$, $n_c$, $S_e$, $S_c$, $X_e$, and $X_c$ are sample sizes, standard deviations, and mean responses in the treatment and control, respectively. lnRR for individual and combined treatments was determined by specifying studies as random-effects factor using the *rma* model in '*metafor*'. The effects of increased/decreased precipitation and warming treatments on soil minerals were considered significant if the 95% CI of lnRR did not overlap zero. We evaluated the relationships between lnRR and precipitation and temperature at the different experimental sites (not precipitation and temperature in the experimental treatments at each particular site) using linear, quadratic, and non-linear (logarithmic, power, and exponential) functions. When two or more functions were significant (p<0.05), we selected the function that maximized $R^2$. Regression analyses were conducted with SigmaPlot 12.5 (Systat Software Inc., San Jose, CA, USA). $P$ values were not adjusted for multiple testing, because this approach can be overly conservative (*Gotelli and Ellison, 2004*).

For the observational studies, we calculated the response of the soil C, N, and P concentrations across precipitation or temperature gradients by estimating the changes in C, N, and P concentrations for each study based on the slopes of the regression lines for the C, N, and P concentrations and precipitation or temperature from a simple linear model. The slopes were then analyzed using a mixed-effects model, with region as a random effect to account for geographic clustering of the study sites. The CI of the resulting grand-mean slope was calculated by parametric bootstrapping

using the *bootMer* function in the 'lme4' package for R (**Bates et al., 2015**) with 1000 simulations and the default parametric bootstrap over both random effects and residual errors for an unconditional CI. We could predict the direction and magnitude of an effect with 95% CI error bars based on the parametric bootstrap. The effects of increased/decreased precipitation and temperature on soil minerals could be considered significant if the 95% CI did not overlap zero.

Parametric bootstrapping across data sets (observational *vs.* experimental studies) was not feasible, so effect size between manipulative experiments and gradient observations could not be formally compared using statistical tests (**Bates et al., 2015**; **Elmendorf et al., 2015**; **Nagy and Grabherr, 2009**). However, even though magnitudes of effects could not be compared statistically, the directions of effects between the two approaches could be compared.

In addition to the meta-analysis explained above, a single multiple regression analysis with a backward stepwise procedure was used to examine the overall patterns of responses of C, N, and P concentrations to multiple climatic (MAT and MAP) and soil variables simultaneously across all the different observational gradients combined. Regression models were developed with increasing numbers of independent variables. The models labelled 'TPASE' included climatic variables (MAT, MAP and aridity), soil classification, and ecosystem as explanatory variables. Overall model significance and goodness-of-fit were judged using the likelihood ratio statistic and assessing changes in Akaike's information criterion scores. Two models were considered different when the change in Akaike's information criterion was >2 for the descriptive ability of the final model over the alternatives (**Chatterjee and Hadi, 2015**). All statistical analyses were performed in R 3.2.4 (R Foundation for Statistical Computing, Vienna, Austria; http://www.R-project.org/, RRID: SCR_001905).

## Acknowledgements

This work was financially supported by The National Key Research and Development Program of China (2016YFA0600801), the National Natural Science Foundation of China (31370455 and 31570438), and One Hundred Person Project of The Chinese Academy of Sciences (K318021405) and of Shaanxi Province. JP and JS are thankful for the support from European Research Council (ERC) Synergy grant ERC-SyG-2013–610028, IMBALANCE-P, Spanish Government grant CGL2016-79835-P, and Catalan Government grant SGR 2014–274. FTM acknowledges support from the ERC Grant Agreements 242658 (BIOCOM) and 647038 (BIODESERT). M.D-B. acknowledges support from the Marie Sklodowska-Curie Actions of the Horizon 2020 Framework Programme H2020-MSCA-IF-2016 under REA grant agreement No. 702057. The funders had no role in the study design, data collection, data analysis, decision to publish, or preparation of the manuscript.

## Additional information

### Funding

| Funder | Grant reference number | Author |
| --- | --- | --- |
| National Key Research and Development Program of China | 2016YFA0600800 | ZY Yuan |
| National Science Foundation of China | 31370455 | ZY Yuan |
| National Science Foundation of China | 31570438 | ZY Yuan |
| European Research Council | ERC-SyG-2013-610028 | Jordi Sardans Josep Peñuelas |
| Spanish Government | CGL2016-79835-P | Jordi Sardans Josep Peñuelas |
| Catalan Government | SGR-2014-274 | Jordi Sardans Josep Peñuelas |
| European Research Council | 242658 | Fernando T Maestre |
| European Research Council | 647038 | Fernando T Maestre |

| Horizon 2020 Framework Programme | Marie Sklodowska-Curie Action 702057 | Manuel Delgado-Baquerizo |
| --- | --- | --- |

The funders had no role in study design, data collection and interpretation, or the decision to submit the work for publication.

## Author contributions

ZYY, Conceptualization, Formal analysis, Funding acquisition, Investigation, Methodology, Writing—original draft, Writing—review and editing; FJ, Data curation, Formal analysis, Investigation, Writing—original draft, Writing—review and editing; XRS, Data curation, Investigation, Methodology, Writing—original draft, Project administration, Writing—review and editing; JS, JP, Writing—original draft, Writing—review and editing; FTM, MD-B, Resources, Writing—original draft, Writing—review and editing; PBR, Conceptualization, Writing—original draft

## Author ORCIDs

ZY Yuan, http://orcid.org/0000-0003-0925-3226
F Jiao, http://orcid.org/0000-0002-3169-4856
XR Shi, http://orcid.org/0000-0003-3963-9269
Fernando T Maestre, http://orcid.org/0000-0002-7434-4856
Josep Peñuelas, http://orcid.org/0000-0002-7215-0150

## Additional files

### Supplementary files

• Supplementary file 1. Supplementary references.

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
