## [Decision Letter]

Thank you for submitting your article "Experimental and observational studies find contrary responses to climate change" for consideration by *eLife*. Your article has been reviewed by three peer reviewers, one of whom, Bernhard Schmid (Reviewer #1) is a member of Board of Reviewing Editors, and the evaluation has been overseen by Ian Baldwin as the Senior Editor. The following individuals involved in review of your submission have agreed to reveal their identity: Hans Lambers (Reviewer #2); Leuzinger Sebastian (Reviewer #3).

The reviewers have discussed the reviews with one another and the Reviewing Editor has drafted this decision to help you prepare a revised submission.

Summary:

This paper provides results from two substantial meta-analyses about effects of global change on soil nutrients. In the first, response ratios of C, N and P to experimentally simulated precipitation, drought or warming from 182 studies are analyzed (E1). In the second, regression slopes of C, N and P in response to 141 naturally occurring gradients in precipitation, drought or temperature are analyzed (O1). Different responses between experimental (E1) and observational studies (O1) are related to short- vs. long-term effects, but other explanations must also be discussed. In further analyses response ratios from experimental studies are plotted as a function of precipitation or temperature at the different studies sites (E2) and the overall response across all observational studies is analyzed as a function of temperature, precipitation, aridity, soil and ecosystem type (O2). If included in the paper, these further analyses must be fully discussed.

Essential revisions:

The reviewers agree that the data and analyses are great but the introduction and especially the interpretation lack precision and depth. These are aspects that can be improved with additional study in a revision.

1) A major point is the comparison between experimental and observational studies. These differ for principal reasons that should be outlined in the introduction, for example referring to similar comparisons in the area of biodiversity-productivity relationships or invasion-biodiversity relationships. Comparisons in these two areas have also sometimes found contrasting responses that could be related to different cause-effects relationships. For the present manuscript, time scale is one among several potential reasons for the differences between observational and experimental studies. For example, we do not know whether experimental studies of much longer duration would lead to response patterns that are more similar to those from observational studies. A caveat of the observational studies used in the meta-analysis is that often they may not have been designed as climate change studies, allowing potential confounding effects to influence results. Perhaps the most striking one would be the change of the vegetation composition along gradients (e.g. if legumes come in this could affect soil N irrespective of precipitation, drought or temperature). This caveat should be mentioned along with the potential confounding effects which might be as important as the difference in time-scale between experimental and observational studies. In this context, it is clear that experimental studies give more consistent results than observational studies.

To better justify that the contrasting results between E1 and O1 may indeed be due to different time-scales and not to the problem of reversed causalities or third variables in observational studies, you should provide focused hypotheses for E1 and O1. Rather than hypothesizing "that different mechanisms are involved", you should suggest those mechanisms and back these suggestions up with references. You should then come back to these mechanisms and discuss them in more detail in the Discussion, using/reviewing for example what the authors of those studies that entered the meta-analysis had suggested.

2) The cross-study analyses E2 and O2 could help to interpret the differences between experimental and observational studies. However, these cross-study analyses have to be better introduced and justified. Again, you should suggest specific hypotheses and then interpret the results along those hypotheses. In the manuscript, you should make it clear when which analyses are being presented: E1 (response ratios), E2 (variation in response ratios across studies), O1 (regression slopes for single gradients), O2 (regressions across studies) all in the Materials and methods section and Results section.

3) Figure 1 should be simplified or omitted. As it stands there are many potential problems with this figure and several arrows could be added or reversed. For example, where an arrow goes from photosynthesis to plant growth, this implies that it is the process of photosynthesis that is affected by temperature and water availability, and that this then affects plant growth. Physiologists have known for a long time that growth is far more sensitive to water availability and temperature than photosynthesis is. Photosynthesis responds, because it is affected by growth (feedback). The way the link is presented between the two processes is fundamentally wrong, but it is also not relevant for your paper to include photosynthesis and stomatal activity in the figure because you do not talk about it. Additionally, many other compartments without further use in the paper could be combined. What do you mean by "Other climates"?

More attention should be given to terminology throughout the paper. For example, the sentence starting in the Discussion section paragraph three tells us that "higher plant productivity" results from "higher accumulation of biomass". Say precisely what you mean by these terms.

4) The title could be more specific by saying "soil nutrient responses" instead of only "responses". The word "contrary" could be replaced by "contrasting", reflecting a more in-depth discussion in the paper of the reasons why it was to be expected that experimental and observational studies should provide different insights into responses to climate change. The paper will have a higher impact if it helps to resolve rather than simply state a contrast. The short running title should better reflect the main work.

[Editors' note: further revisions were requested prior to acceptance, as described below.]

Thank you for resubmitting your work entitled "Experimental and observational studies find contrasting soil nutrient responses to climate change" for further consideration at *eLife*. Your revised article has been evaluated by Ian Baldwin (Senior editor) and Bernhard Schmid (Reviewing editor).

The manuscript has been improved but there are some remaining issues that need to be addressed before acceptance, as outlined below:

First of all, we are pleased that you could use our suggestions in this revision. In particular, it is important that you have added additional reasons for differences between experimental and observational studies and that you provided more explanations for the predicted and observed effects between climate change and soil nutrients in the Introduction and Discussion, respectively.

There is one major point that we are still struggling with, and, apparently, some of your co-authors do as well. It is what we referred to previously as the across-study analyses E2 and O2. It is not clear how you compare them in the paragraphs that are marked yellow in the main text and in the many supplements (not marked). E2 analyzes effects of variation between experimental studies in site temperature and precipitation on response ratios of nutrient concentrations found between treatment and control within the different individual studies. In contrast, O2 analyzes effects of between – plus within-study variation in site temperature and precipitation on values of nutrient concentrations. It would be possible to use the sites of the experimental studies as an observational gradient and analyze values of nutrient concentrations found at these sites, without reference to the experiments. I am not sure if this was done, but it would probably not make much sense. Another, more sensible possibility, which was not done, would be to analyze the variation between observational studies in site temperature and precipitation on slopes of nutrient concentrations against climatic variables within the different individual studies. This would then be comparable with E2.

Indeed, you should also add the legend words response ratios and slopes to your Figure 2–Figure 4 (as you did in Figure 2—figure supplement 1). Even though these are different, they can be compared in a meta-analysis of meta-data (a meta-meta-analysis). But you don't have to do this because it can also be done visually with Figure 2-4.

Relating to the previous two paragraphs, we suggest that you remove most of your supplementary material because it does not add to your main message and only confuses those who might actually look at this material. What is much more important is that you provide the data in an easily accessible way so that interested readers can recreate themselves such material as you wanted to present in the supplementary material.

Another point that you must clarify is how your two own observational data sets are contributing to the analysis. Are these two studies two of the 141 observational ones? If yes, say so, if no, explain how many studies the two studies represented of the 141 or if they were actually added to the 141. You could also consider to use particular symbols for these data sets in the figures.

[Editors' note: further revisions were requested prior to acceptance, as described below.]

Thank you for resubmitting your work entitled "Experimental and observational studies find contrasting soil nutrient responses to climate change" for further consideration at *eLife*. Your revised article has been favorably evaluated by Ian Baldwin (Senior editor) and Bernhard Schmid (Reviewing editor).

The manuscript has been improved but there are some remaining issues that need to be addressed before acceptance, as outlined in the track changes and comments in the attached version. We agree that it is not necessary anymore to analyze the slopes as a function of mean climatic variables of the observational studies. Instead it is fine to simply present the overall multiple regressions. It is also not necessary to show Spearman rank correlation results and we have deleted those parts. However, we think it makes sense to add Figure 7 because it actually strengthens your case, in particular because it shows the declines of C and N with MAT better than does the overall analysis of the observational studies.

Please make sure you check if the aridity index in Figure 6 and Figure 7 is the same as the aridity gradient used in Figure 3. We think this is not the case and very confusing. If necessary, you should "reverse" the x-axis in Figure 6–Figure 7 by using the same measure as in Figure 3. Like this the soil nutrients will be negatively correlated with decreased water supply as they should (you write in the Results "Soil N and P concentrations decreased with increasing aridity…").

---

## [Author Response]

Essential revisions:

The reviewers agree that the data and analyses are great but the introduction and especially the interpretation lack precision and depth. These are aspects that can be improved with additional study in a revision.

1) A major point is the comparison between experimental and observational studies. These differ for principal reasons that should be outlined in the introduction, for example referring to similar comparisons in the area of biodiversity-productivity relationships or invasion-biodiversity relationships. Comparisons in these two areas have also sometimes found contrasting responses that could be related to different cause-effects relationships. For the present manuscript, time scale is one among several potential reasons for the differences between observational and experimental studies. For example, we do not know whether experimental studies of much longer duration would lead to response patterns that are more similar to those from observational studies. A caveat of the observational studies used in the meta-analysis is that often they may not have been designed as climate change studies, allowing potential confounding effects to influence results. Perhaps the most striking one would be the change of the vegetation composition along gradients (e.g. if legumes come in this could affect soil N irrespective of precipitation, drought or temperature). This caveat should be mentioned along with the potential confounding effects which might be as important as the difference in time-scale between experimental and observational studies. In this context, it is clear that experimental studies give more consistent results than observational studies.

We acknowledge these comments, and have expanded the Introduction and Discussion sections to accommodate them.

To better justify that the contrasting results between E1 and O1 may indeed be due to different time-scales and not to the problem of reversed causalities or third variables in observational studies, you should provide focused hypotheses for E1 and O1. Rather than hypothesizing "that different mechanisms are involved", you should suggest those mechanisms and back these suggestions up with references. You should then come back to these mechanisms and discuss them in more detail in the Discussion, using/reviewing for example what the authors of those studies that entered the meta-analysis had suggested.

In this revision, we have included your suggestions in the sections of Introduction (paragraph two) and Discussion (paragraph seven).

2) The cross-study analyses E2 and O2 could help to interpret the differences between experimental and observational studies. However, these cross-study analyses have to be better introduced and justified. Again, you should suggest specific hypotheses and then interpret the results along those hypotheses. In the manuscript, you should make it clear when which analyses are being presented: E1 (response ratios), E2 (variation in response ratios across studies), O1 (regression slopes for single gradients), O2 (regressions across studies) all in the Materials and methods section and Results section.

According to your suggestions, we have made them clear in the sections of Introduction, Material & methods and Results.

3) Figure 1 should be simplified or omitted. As it stands there are many potential problems with this figure and several arrows could be added or reversed. For example, where an arrow goes from photosynthesis to plant growth, this implies that it is the process of photosynthesis that is affected by temperature and water availability, and that this then affects plant growth. Physiologists have known for a long time that growth is far more sensitive to water availability and temperature than photosynthesis is. Photosynthesis responds, because it is affected by growth (feedback). The way the link is presented between the two processes is fundamentally wrong, but it is also not relevant for your paper to include photosynthesis and stomatal activity in the figure because you do not talk about it. Additionally, many other compartments without further use in the paper could be combined. What do you mean by "Other climates"?

More attention should be given to terminology throughout the paper. For example, the sentence starting in the Discussion section paragraph three tells us that "higher plant productivity" results from "higher accumulation of biomass". Say precisely what you mean by these terms.

This figure has been removed based on your suggestion.

4) The title could be more specific by saying "soil nutrient responses" instead of only "responses". The word "contrary" could be replaced by "contrasting", reflecting a more in-depth discussion in the paper of the reasons why it was to be expected that experimental and observational studies should provide different insights into responses to climate change. The paper will have a higher impact if it helps to resolve rather than simply state a contrast. The short running title should better reflect the main work.

As you mentioned for the two approaches, neither is a perfect surrogate for climate change. We cannot say which is right or wrong. Different approaches just tell us different things and we need to do a better job of laying those out, e.g. when and where each approach may be informative. Experiments may tell us short-term responses (years to decades) prior to coincident shifts in plant or microbial composition which may counteract the short-term responses; so these may be better for predicting responses given our current plants and soils; observational gradients may better tell us what happens over a longer time frame when multiple aspects of the system have had a chance to adjust. According to your suggestion, we have changed the title and added more discussion (Discussion paragraphs five and seven) in this revised version.

[Editors' note: further revisions were requested prior to acceptance, as described below.]

[…]

There is one major point that we are still struggling with, and, apparently, some of your co-authors do as well. It is what we referred to previously as the across-study analyses E2 and O2. It is not clear how you compare them in the paragraphs that are marked yellow in the main text and in the many supplements (not marked).

We have re-organized the text and clarified it accordingly in this revised version.

E2 analyzes effects of variation between experimental studies in site temperature and precipitation on response ratios of nutrient concentrations found between treatment and control within the different individual studies. In contrast, O2 analyzes effects of between – plus within-study variation in site temperature and precipitation on values of nutrient concentrations. It would be possible to use the sites of the experimental studies as an observational gradient and analyze values of nutrient concentrations found at these sites, without reference to the experiments. I am not sure if this was done, but it would probably not make much sense.

Using the controls from experiments should lead to similar conclusions that we reached using the observational dataset, but with a lower resolution (due to lower number of samples). According to your suggestion, we have now added a new Figure by repeating observational analyses but only using the control from all experiments instead of all observational data (see Results paragraph four; p29, Figure 7).

In addition, as you mentioned, E2 analyzes the changes in response rations whereas O2 analyzes nutrients with climate variables. So the slopes of E2 and O2 are different things. We therefore have now removed Table 3.

Another, more sensible possibility, which was not done, would be to analyze the variation between observational studies in site temperature and precipitation on slopes of nutrient concentrations against climatic variables within the different individual studies. This would then be comparable with E2.

This point is unclear for us. We are not sure if you want us to study slopes in observations of each site, i.e., calculate the slope of nutrient concentration relative to temperature and precipitation range within each site, and then plot these slopes versus the corresponding temperature and precipitation of each site. However, for spatially observational studies, there is only one observation at each site, i.e., only one *x* value for each *y* value at each site. Therefore, it is impossible to calculate slope at each site. Similarly, we can't get a slope at each site for experimental studies. We have instead calculated the slopes for different individual observational studies (not for each site) along the corresponding environmental gradients. That is to say, there is one slope for one gradient observation along several climate data for one study. We got various slopes from different individual studies and used *bootMer* in *lme4* package to get a grand-mean slope, as shown in Figure 2–Figure 4. The overall slope, as shown in Figure 6, can be drawn by using all data from all studies for all sites. We are not sure if you mean that the slope of the relationship between climate and nutrients does not work for quadratic relationships. In that case, we have calculated the Spearman correlation between climate and nutrients in order to reduce your concern. We will be happy to test any other relationship you had in mind for us not fully test.

Indeed, you should also add the legend words response ratios and slopes to your Figure 2–Figure 4 (as you did in Figure 2—figure supplement 1). Even though these are different, they can be compared in a meta-analysis of meta-data (a meta-meta-analysis). But you don't have to do this because it can also be done visually with Figure 2–Figure 4.

We have added these legend words accordingly.

Relating to the previous two paragraphs, we suggest that you remove most of your supplementary material because it does not add to your main message and only confuses those who might actually look at this material. What is much more important is that you provide the data in an easily accessible way so that interested readers can recreate themselves such material as you wanted to present in the supplementary material.

According to your suggestion, we have removed most of these supplementary material, especially the supplementary tables and figures that may confuse readers.

Another point that you must clarify is how your two own observational data sets are contributing to the analysis. Are these two studies two of the 141 observational ones? If yes, say so, if no, explain how many studies the two studies represented of the 141 or if they were actually added to the 141. You could also consider to use particular symbols for these data sets in the figures.

According to your suggestion, we have clarified that the 141 observations included our two data sets. Also, in this new version, we have used particular symbols for our data sets (see Figure 1).

[Editors' note: further revisions were requested prior to acceptance, as described below.]

The manuscript has been improved but there are some remaining issues that need to be addressed before acceptance, as outlined in the track changes and comments in the attached version. We agree that it is not necessary anymore to analyze the slopes as a function of mean climatic variables of the observational studies. Instead it is fine to simply present the overall multiple regressions. It is also not necessary to show Spearman rank correlation results and we have deleted those parts. However, we think it makes sense to add Figure 7 because it actually strengthens your case, in particular because it shows the declines of C and N with MAT better than does the overall analysis of the observational studies.

According to your suggestion, we have removed that table and kept that figure as you mentioned above.

Please make sure you check if the aridity index in Figure 6 and Figure 7 is the same as the aridity gradient used in Figure 3. We think this is not the case and very confusing. If necessary, you should "reverse" the x-axis in Figure 6–Figure 7 by using the same measure as in Figure 3. Like this the soil nutrients will be negatively correlated with decreased water supply as they should (you write in the Results "Soil N and P concentrations decreased with increasing aridity…").

According to your suggestion, we have re-drawn the two figures by changing aridity index to aridity (see revised Figure 6 and Figure 7). The related text has also been revised accordingly (Results paragraph four).